# Water Quality Responses during the Continuous Mixing Process and Informed Management of a Stratified Drinking Water Reservoir

**Zizhen Zhou [1],\*, Tinglin Huang [2],\*, Weijin Gong [1], Yang Li [1], Yue Liu [1], Shilei Zhou [3] and Meiying Cao [4]**

[1] School of Energy and Environment, Zhongyuan University of Technology, Zhengzhou 450007, China; 5536@zut.edu.cn (W.G.); ly_zut@163.com (Y.L.); yue5757@sina.com (Y.L.)

[2] School of Environmental and Municipal Engineering, Xi'an University of Architecture and Technology, Xi'an 710055, China

[3] School of Environment Science and Engineering, Hebei University of Science and Technology, Shijiazhuang 050018, China; zslzhoushilei@126.com

[4] Shangqiu Kangda Sewage Treatment Co. LTD, Shangqiu 476000, China; MeiyingCao@163.com

\* Correspondence: 6623@zut.edu.cn (Z.Z.); huangtinglin@xauat.edu.cn (T.H.);
Tel.: +86-0371-62506813 (Z.Z.); +86-029-82201038 (T.H.)

**Abstract:** Aeration and mixing have been proven as effective in situ water quality improvement methods, particularly for deep drinking water reservoirs. While there is some research on the mechanism of water quality improvement during artificial mixing, the changes to water quality and the microbial community during the subsequent continuous mixing process is little understood. In this study, we investigate the mechanism of water quality improvement during the continuous mixing process in a drinking water reservoir. During this period, we found a reduction in total nitrogen (TN), total phosphorus (TP), ammonium-nitrogen ($NH_4$-N), iron (Fe), manganese (Mn), and total organic carbon (TOC) of 12.5%–30.8%. We also measured reductions of 8.6% and 6.2% in TN and organic carbon (OC), respectively, in surface sediment. Microbial metabolic activity, abundance, and carbon source utilization were also improved. Redundancy analysis indicated that temperature and dissolved oxygen (DO) were key factors affecting changes in the microbial community. With intervention, the water temperature during continuous mixing was 15 °C, and the mixing temperature in the reservoir increased by 5 °C compared with natural mixing. Our research shows that integrating and optimizing the artificial and continuous mixing processes influences energy savings. This research provides a theoretical basis for further advancing treatment optimizations for a drinking water supply.

**Keywords:** continuous mixing; water quality improvement; microbial community; pollutant removal

## 1. Introduction

There are an increasing number of reservoirs being used as drinking water sources for humans. However, although reservoirs are an important drinking water source, they are subjected to considerable water quality problems such as eutrophication and oxygen depletion [1]. Reservoir operation strategies affect biogeochemical processes associated with the cycling of nutrients, particularly nitrogen and organic matter [2,3]. Water quality problems have often been associated with algal overproduction, leaching of pollutants from sediments, and incoming storm water runoff [4]. These mechanisms may seriously compromise the safety of a water supply. Of all the water quality indicators, temperature is a critical parameter that influences aquatic microbes and material transfers. Temperature regulates chemical reaction rates and affects microbial community distribution [5]. Dissolved oxygen (DO) is also

important for microbial metabolism and controlling redox reactions [6]. During the stratification period, variations in water temperature and DO concentrations lead to a series of water quality problems. Thus, the removal of nitrogen and organic matter from drinking water reservoirs is gaining research attention [7,8]. For example, one of the major concerns about the production of drinking water is represented by disinfectant by-products (DBPs), which can appear after using disinfectants to treat water with a high concentration of organic compounds [9–11].

Of all the technologies, including physical, chemical, and biological technologies, artificial mixing proved to be effective and free from secondary pollution [12]. Water lifting aeration systems have been installed into some reservoirs for water quality improvement [13,14]. Satisfying results have been achieved in preventing stratification, oxygenating water, and removing pollutants, but there has been no further optimization in the energy consumption of water lifting aeration systems. In fact, natural mixing of water may also supplement oxygen and eliminate stratification of the reservoir [15]. Energy may be saved if we take water lifting aeration systems as an inducing mixing tool and make full use of natural mixing conditions. However, there is little research on water quality processes in the reservoir during the continuous mixing period after the artificially induced mixing period. Additionally, research on the transfer and transformation of microbes and pollutants is scant.

Deep lakes or reservoirs (depth >10 m) experience a prolonged stratification period in the summer. Most water quality problems occur during this period. In the Heihe Jinpen Reservoir, the stratification period is from April to October. During this time, pollutants from sediments are released into the lower water column. After artificial mixing, the water quality of the reservoir was improved to some extent [16]. However, understanding regarding how the reservoir water quality changes, including the leaching of pollutants from sediments, during the continuous mixing period is not clear. This study investigates the reasons for these changes. The purpose of this research was to: (1) During the continuous mixing period, analyze the responses of TN, TP, TOC, Fe, and Mn concentrations in the reservoir, (2) use the Biolog ECOplate to explore microbial metabolic activity and its utilization of carbon sources in the reservoir, (3) explore variations associated with nitrogen, organic matter, and metals in the microbial community, and (4) determine the connections between artificial and continuous mixing.

## 2. Materials and Methods

### 2.1. Reservoir for Field Research and Sampling Methods

Our study focused on the Heihe Jinpen Reservoir (34°42′–34°13′ N; 107°43′–108°24′ E), located in Zhouzhi County, Xi'an City, Shaanxi Province, a drinking water reservoir in Xi'an City. This reservoir has a capacity of 200 million $m^3$ with an area of 3 $km^2$. As shown in Figure 1, three sampling sites denoted S1, S2, and S3 were chosen in the main reservoir. Three parallel water samples, including surface water (0.5 m below the water surface), middle water, and bottom water (0.5 m above the reservoir bed) were collected [17]. As the water column at the time of sampling was considered to be evenly mixed, the vertical water quality tended to be the same; therefore, the pollutant concentrations used in this research were the mean values in the water samples. After collection, the samples were quickly put into a refrigerator at 4 °C for analysis.

In terms of sediment samples from the three sites, 0.5 cm was collected from the uppermost layer using a stratified sediment sampler [4]. All samples were refrigerated and dried for analysis and determination.

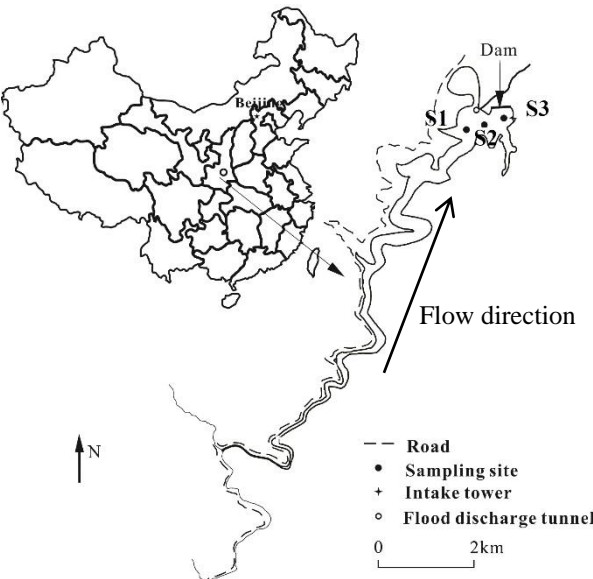

**Figure 1.** Sampling sites in the Heihe Jinpen Reservoir.

## 2.2. Determination Methods of Physicochemical Index

Temperature and DO were determined in situ using a Hach DS5 instrument (Loveland, CO, USA). Total nitrogen (TN) and ammonium-nitrogen ($NH_4$-N) concentrations were determined using a SEAL AA3 HR auto analyzer (SEAL, Hamburg, Germany). Iron (Fe) and manganese (Mn) concentrations were measured by inductively coupled plasma mass spectrometry (ICP-MS; Thermo, Waltham, MA, USA). Total organic carbon (TOC) was determined by TOC analyzer (Shimadzu, Kyoto, Japan) [16].

The organic matter content in the sediment was estimated as weight loss by ignition (LOI %) at 550 °C for 5 h [4]. To measure Fe and Mn, sediment samples were digested with a $HCl–HNO_3–HF–HClO_4$ mixture and then analyzed via ICP-MS (Thermo, Waltham, MA, USA) [18]. The TN concentration in the sediments was determined using the basic potassium persulfate method [19] and the TP concentration was determined using the standard measurements and testing (SMT) method for freshwater sediment developed under the framework of the European Standards and Testing Commission [20].

## 2.3. Determination Methods of Microbial Metabolism

The Biolog ECOplate was part of the American Gen III MicroStation automatic microbial identification instruments used in this study [21,22]. The 96 carbon source holes (3 blank control holes) of the ECOplate were cultured in a 28 °C incubator, and the absorbance values at 590 nm and 750 nm were measured every 24 h. The average well color development (AWCD) index reflected the total metabolic activity of microorganisms given different carbon sources. AWCD values = Σ (C590-750)/31 [23]. The carbon source utilization value (dimensionless) was the average absorbance value of each carbon source. The Shannon and McIntosh indices were also calculated from the absorbances [24].

## 2.4. High-Through Sequencing

The sequencing method was the same as our previous research. DNA extraction and Illumina MiSeq High-throughput sequencing were carried out by Meiji biology Co., Ltd. (Shanghai, China) [16].

### 2.5. Data Analysis

Measurement data were shown as mean ± standard deviation. We used the software SPSS 23.0 to conduct one way analysis of variance (ANOVA) to determine the statistical significance ($p < 0.05$) for the environmental and microbial community parameters.

## 3. Results and Discussion

### 3.1. Continuous Mixing Process

In our previous study, aeration mixing played a key role in improving water quality and inducing variation in the microbial community [16,22]. The results of the operation of the water lifting aeration system are shown in Figure S1 (Supplementary Materials). After the operation of the water-lifting aerators, the water quality in the Heihe Jinpen Reservoir remained in good condition from November to March (four months) compared with the quality from April to October. Figure 2 (mean values of S1, S2, and S3) illustrates how, after the use of the water-lifting aerators, the difference between bottom water and surface water temperatures decreased to less than 3 °C, and the DO concentration in the bottom water increased to more than 2 mg/L (on the first day after operation). Upon continuation of the mixing process, the water temperature differential fell to less than 1 °C, and DO continued to rise to nearly 10 mg/L in the water column. This meant that the reservoir water experienced natural mixing, due to changes in the season and air temperature, after the system stopped running. After the 9th day, the water column became oxygen-rich.

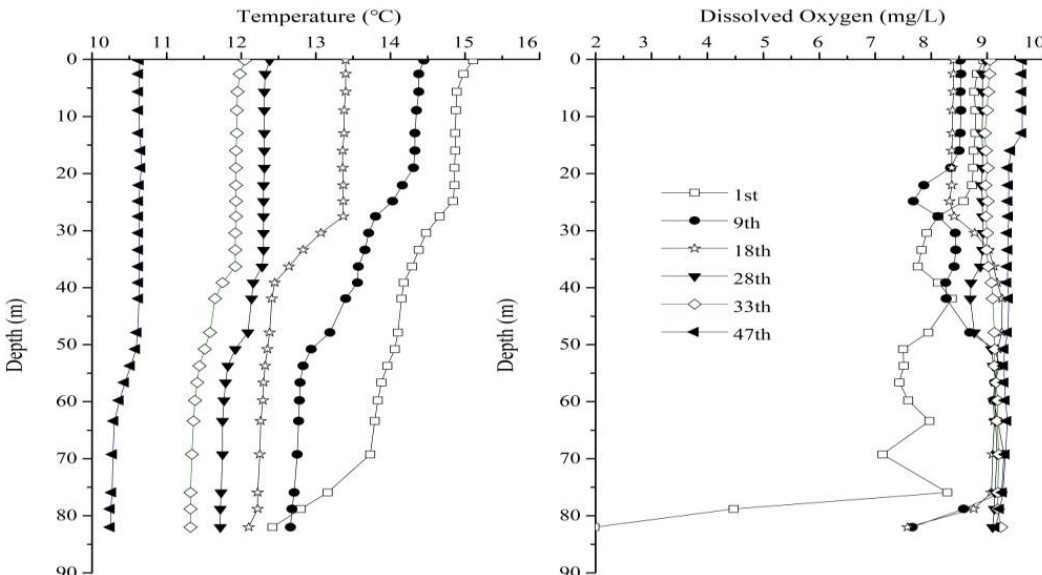

**Figure 2.** Water temperature and dissolved oxygen (DO) concentration variations in the water column during the continuous mixing process (9th means 9 days since the lifting aeration system was stopped).

The depletion of DO in the water column directly led to the deterioration of water quality [25]. Stratification and vertical mixing have previously been identified as key factors that cause phytoplankton overproduction [26]. Thus, destratification and improving DO concentration in the reservoir are effective solutions to water quality problems. In our study, continuous natural oxygenation and destratification have been observed in the continuous process. Compared with artificial mixing and oxygenation, this process saves a substantial amount of energy. Additionally, we observed ongoing improvements to key water quality indicators.

## 3.2. Changes to Key Water Quality Indicators during the Continuous Mixing Process

During the natural mixing process, water temperature decreased to nearly 10 °C and DO increased to 10 mg/L all along the water column. At the same time, the mean values of TN, TP, $NH_4$-N, Fe, Mn, and TOC of the vertical water column showed a decreasing trend as detailed in Table 1. This meant that during the continuous mixing process, the water quality in the Heihe Jinpen Reservoir continued to improve. On the first day after operation, TN was 1.12 ± 0.05 mg/L and decreased to 0.98 ± 0.02 mg/L on day 47; a reduction of 12.5%. TP concentration on the first day after operation was 0.025 ± 0.01 mg/L; this decreased to 0.020 ± 0.005 mg/L on day 47, which is a reduction of 20.0%. The $NH_4$-N concentration decreased from 0.45 ± 0.01 mg/L to 0.33 ± 0.02 mg/L, a reduction of 26.7%. The Fe concentration decreased from 0.22 ± 0.01 mg/L to 0.18 ± 0.03 mg/L, falling by 18.2%. The Mn concentration decreased from 0.13 ± 0.02 mg/L to 0.09 ± 0.01 mg/L, a reduction of 30.8%. The TOC concentration decreased from 2.65 ± 0.22 mg/L to 2.32 ± 0.15 mg/L, a drop of 12.5%.

**Table 1.** Changes to the key water quality indicators during the continuous mixing process in the water column.

| Water Quality Indicators/Sampling Date | TN (mg/L) | TP (mg/L) | $NH_4$-N (mg/L) | Fe (mg/L) | Mn (mg/L) | TOC (mg/L) |
|---|---|---|---|---|---|---|
| 1st | 1.12 ± 0.05 | 0.025 ± 0.01 | 0.45 ± 0.01 | 0.22 ± 0.01 | 0.13 ± 0.02 | 2.65 ± 0.22 |
| 9th | 1.08 ± 0.08 | 0.022 ± 0.008 | 0.32 ± 0.03 | 0.22 ± 0.02 | 0.13 ± 0.02 | 2.55 ± 0.25 |
| 18th | 1.10 ± 0.05 | 0.022 ± 0.008 | 0.39 ± 0.05 | 0.20 ± 0.05 | 0.15 ± 0.01 | 2.62 ± 0.20 |
| 28th | 1.15 ± 0.05 | 0.027 ± 0.01 | 0.42 ± 0.05 | 0.25 ± 0.03 | 0.12 ± 0.02 | 2.40 ± 0.20 |
| 33rd | 1.05 ± 0.1 | 0.020 ± 0.005 | 0.28 ± 0.01 | 0.18 ± 0.02 | 0.11 ± 0.03 | 2.33 ± 0.20 |
| 47th | 0.98 ± 0.02 | 0.020 ± 0.005 | 0.33 ± 0.02 | 0.18 ± 0.03 | 0.09 ± 0.01 | 2.32 ± 0.15 |

Note: TN: total nitrogen; TP: total phosphorus; $NH_4$-N: ammonium-nitrogen; Fe: iron; Mn: manganese; TOC: total organic carbon.

Natural mixing did not result in the same magnitude of pollutant reduction compared with artificial mixing. Macrophyte absorption, particulate matter sedimentation [27], and activation of the indigenous microbial community are methods frequently used for nutrient removal [28]. Another study also suggests that decreased water temperature aids in the reduction of pollutant leaching from the reservoir sediment [29]. As such, during this mixing period, decreasing water temperatures may also contribute to a reduction in nutrient concentrations. In our previous research on this reservoir, we observed an increase in the number of indigenous aerobic denitrifying bacteria that corresponded with a reduction in the concentration of N during the artificial mixing process. We were also interested in understanding the changes to the microbial community during natural mixing. Microbial experiments were carried out during the natural mixing period based on the water quality results.

## 3.3. Microbial Metabolism Variations during Continuous Mixing Process

The AWCD results of the Biolog ECOplate test is shown in Figure 3. During the continuous mixing process, with the decrease of water temperature, the AWCD values of the water body increased. This meant that microbial metabolic activities within the water body increased. The 144 h of cultivation was typically excluded as an example for the explanation. We can see that, on the first day, the AWCD value was 0.657 and then gradually increased to 0.827 on day 47, increasing by 25.9%. At 240 h of cultivation, the AWCD values increased from 0.793 on day 1 to 0.930 on day 47.

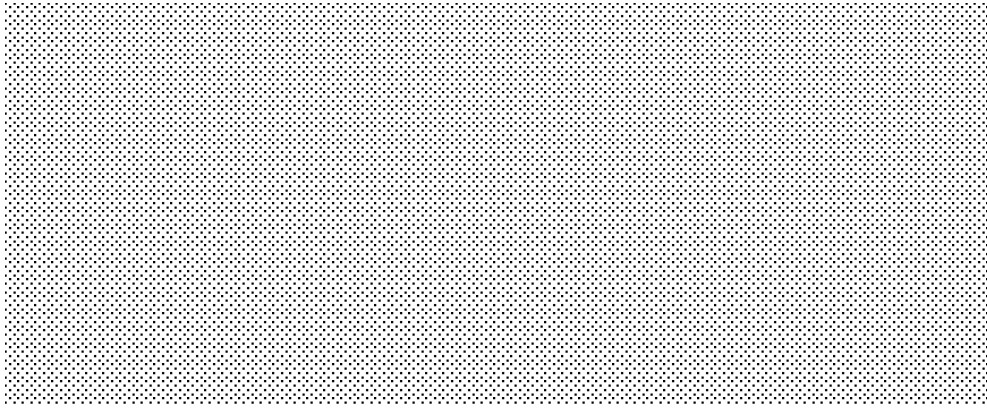

**Figure 3.** Average well color development (AWCD) variations in the water column during the continuous mixing process (9th means 9 days since the lifting aeration system was stopped).

The Heihe Jinpen Reservoir is a drinking water reservoir and, as such, the AWCD values are lower than those for organic wastewater. Organic wastewater AWCD values can be as high 1.20, with different microbial species exhibiting different metabolic activities [30]. This suggests that the metabolic activities of microbial communities also increase, given sufficient DO.

A previous study showed that the carbon sources utilization pattern changes due to the adaptation and selective growth of the microbial population [31], and a similar pattern was also seen in our research. Figure 4a shows that, generally, the total utilization of carbon sources by microbes increased gradually. On day 1, the total utilization of six carbon sources was 4.31; this increased to 5.093 on day 18 and finally to 5.010 on day 47. This trend indicates that the utilization of carbon sources by microbes increased during the continuous mixing period. As for carboxylic acid substrate, utilization increased from 0.543 on day 1 to 0.797 on day 47, an increase of 46.8%. Conversely, for amine, utilization showed a decreasing trend from 0.810 to 0.783.

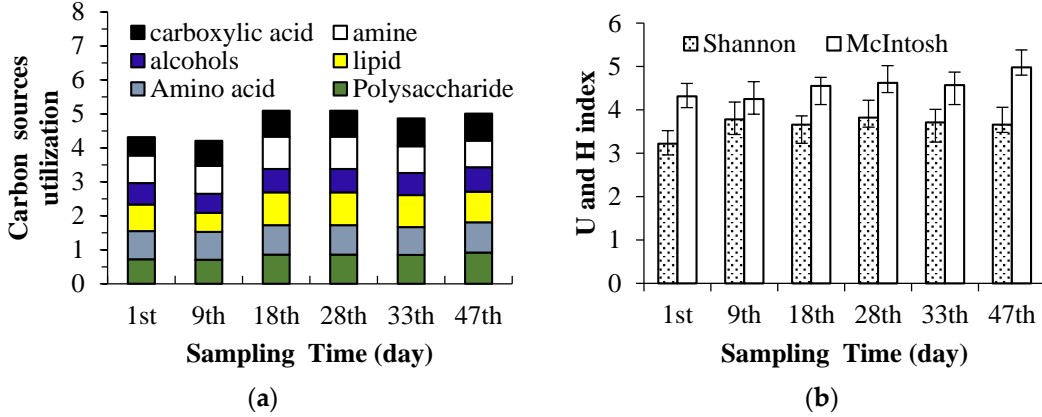

**Figure 4.** Carbon sources utilization (**a**) and abundances of microorganisms (**b**) during the continuous mixing process (9th means 9 days since the lifting aeration system was stopped).

The Shannon index (U index) and McIntosh index (H index) are measures of the diversity of microbial functions. Figure 4b shows that, even though water temperature decreased, the total microbial abundance increased slightly. The U index increased from 3.22 on day 1 to 3.82 on day 28, and finally to 3.66 on day 47. The H index increased from 4.31 on day 1 to 4.98 on day 47. The U index and H index increased by 13.66% and 15.55%, respectively.

Polysaccharides, amino acids, lipids, alcohols, amines, and carboxylic acids are part of the TOC. *Actinomycete*, which was found in the Heihe Jinpen Reservoir, is part of the water organic matter cycle [32]. *Bacteroidetes* is also important for the removal of dissolved organic matter. This shows that these bacteria metabolized part of the carbon sources and then reduced the TOC in the reservoir.

*Acidovorax, Rhizobium*, and *Sphingomonas* [33], which were also found in the Heihe Jinpen Reservoir, were affected mainly by DO. This gives us some insights into the nitrogen removal mechanisms at play and shows that the improvement of microbial metabolism (AWCD value) helps in removing nitrogen from water. Sufficient DO in water helps to control the concentration of Fe and Mn and oxidizes some reducing substances [34]. This plays an important role in water purification. A greater abundance and higher metabolic activity (U and H indices) of microbial communities influenced the circulation of pollutants in the reservoir and sediments [35,36]. Based on the above results, high microbial metabolic efficiency also played an important role in nutrient removal.

### 3.4. Microbial Community Structure Variations during the Continuous Mixing Process

Sequencing results indicated that the microbial community structure also changed during the continuous mixing process. As shown in Figure 5, on a phylum level, the percentage of *Proteobacteria* increased from 36.1% on day 1 to 44.1% on day 47. The percentage of *Actinobacteria* on day 1 was 24.8% and increased to 28.6% on day 47. The percentage of *Bacteroidetes* varied from 10.4% to 15.1%, and that of *Firmicutes* varied from 1.8% to 12.1%. The total number of phyla detected also increased from 33,431 to 45,379.

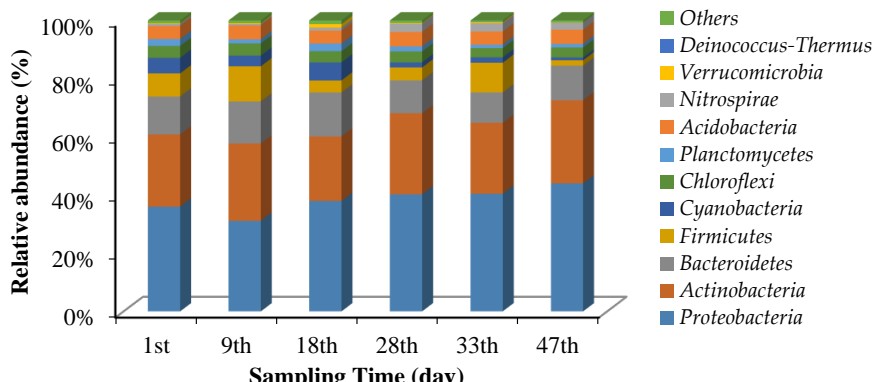

**Figure 5.** Microbial community structure variation on phylum level during the continuous mixing process.

Microbial community structure analysis results on genus level are shown in Figure 6. The main genus was *hgcI_clade*, and its content increased from 16.3% on day 1 to 19.5% on day 47. The percentage of *LD12_freshwater_group_norank* ranged from 6.9% on day 1 to 14.6% on day 47. *Cyanobacteria_norank* showed a decreasing trend from 5.3% on day 1 to 0.7% on day 47. *LD28_freshwater_group* showed an increasing trend from 4.7% on day 1 to 9.2% on day 47.

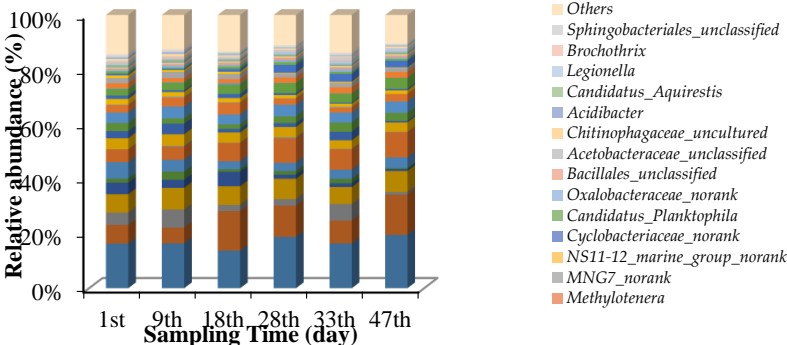

**Figure 6.** Microbial community structure variation on genus level during the continuous mixing process.

These results showed that, during this natural process, the microbial community structure (both on phylum and genus levels) varied significantly despite there being little interference to the reservoir.

*hgcI_clade*, often found in shallow lakes, accounted for the largest proportion of the community and showed distinct metabolic and ecological functions. *Acinetobacter*, which has a better $NH_4$-N removal capacity, was identified as the main microbial community in the Heihe Jinpen Reservoir [37]. Oxytetracycline, a type of organic matter, can be degraded by *Pseudomonas* in the presence of $Fe^{3+}$ [38]. Other research also found that Fe removal from raw asbestos by siderophores-producing *Pseudomonas* was feasible [39]. Given these examples, many microbial communities may prove to influence the fate of organic matter, nitrogen, and metal pollutants during the continuous mixing period. In a previous study [22], we already found that some functional bacteria were activated and helped to remove nitrogen and TOC in the reservoir during the artificial mixing period. Thus, the self-purification capacity of the water remained high during this natural period. This was because natural mixing had been advanced by the artificial mixing process (which required an energy input). At this time, a high air temperature kept the water body at a high temperature as well. Appropriate water temperature and high DO concentrations are thus key factors to maintaining the high metabolic activity of microbes in the reservoir.

### 3.5. The Fate of Pollutants

We also investigated the migration and transformation of the pollutants studied in this research. During the continuous mixing process, the pollutant concentrations in the surface sediments were determined. As shown in Table 2, TN in sediment showed a decreasing trend from 1666 ± 156 mg/kg to 1523 ± 142 mg/kg. This variation was nearly the same as those in the water column, indicating that nitrogen, both in water and sediment, were removed by bacteria involved in the nitrogen cycle. This phenomenon has also been confirmed by the literature [4]. TP concentrations in sediment remained at 800–900 mk/kg, because the phosphorus was an inert element. Even with the action of microbes, it will not be removed from the reservoir. At most, it may only migrate from the water into the surface sediment. Fe and Mn in sediments showed an increasing trend, indicating that they were oxidized into a stable state by sufficient DO in the water column and then deposited onto the surface sediment [40].

**Table 2.** Changes of pollutant indictors during the continuous mixing process in the surface sediment.

| Pollutant Indicators/ Sampling Date | TN (mg/kg) | TP (mg/kg) | Fe (mg/kg) | Mn (mg/kg) | OC (%) |
|---|---|---|---|---|---|
| 1st | 1666 ± 156 | 856 ± 85 | 16244 ± 1140 | 142 ± 12 | 3.21 ± 0.32 |
| 9th | 1578 ± 142 | 889 ± 52 | 15740 ± 1024 | 137 ± 13 | 3.02 ± 0.21 |
| 18th | 1522 ± 133 | 792 ± 121 | 16890 ± 885 | 118 ± 9 | 2.98 ± 0.20 |
| 28th | 1600 ± 160 | 901 ± 98 | 16021 ± 741 | 141 ± 8 | 3.08 ± 0.33 |
| 33rd | 1521 ± 172 | 779 ± 102 | 16444 ± 123 | 132 ± 15 | 2.99 ± 0.15 |
| 47th | 1523 ± 142 | 877 ± 69 | 16779 ± 556 | 156 ± 14 | 3.01 ± 0.18 |

OC in sediment showed a slightly decreasing trend from 3.21% ± 0.32% to 3.01% ± 0.18%. The higher microbial metabolic activity may have caused this reduction [41].

### 3.6. The Relationship between Environmental Factors and the Microbial Community

Redundancy analysis (RDA) was used to analyze the relationship between water quality and the microbial community. According to the RDA analysis at phylum and genus levels, shown in Table S1 (Supplementary Materials), physical factors (temperature, DO) and chemical factors (Chl-a (Chlorophyll-a), TOC, TN) were key in affecting change in the microbial community in the reservoir. During the continuous mixing period, these physicochemical factors changed, thus impacting on the water quality.

On the phylum level, temperature and CHl-a were the most important drivers that exhibited significant positive correlation with RDA1, and the correlation coefficient reached 0.76($p < 0.05$) and 0.53 ($p < 0.05$). On the genus level, TN (R = 0.42, $p < 0.05$), TOC (R = 0.47, $p < 0.05$), temperature (R

= 0.64, $p < 0.05$), DO (R = $-0.40$, $p < 0.05$), and CHl-a (R = 0.54, $p < 0.05$) were key environmental factors that exhibited significant correlation with RDA1. From the above results, we can see that, during the continuous mixing period, the most important physical factors of water temperature and DO concentrations were changed significantly. For the improvement of water quality, we should focus on the control of water temperature and DO.

## 4. Conclusions

Based on this study and our previous research, we could conclude that aeration and mixing can destroy the stratification of the reservoir and activate the metabolism ability and abundance of indigenous microbes in the reservoir. The water purification mechanism has also been confirmed. When the water quality at the bottom of the reservoir deteriorated and surface algae bloomed, the aeration system started running. When the water quality index reached a pre-determined target, the system stopped running. This is the typical practice in this field of research.

This study demonstrates a new phenomenon, showing that there are improvements to water quality during the continuous mixing process. This study advocates the integration of artificial mixing with natural mixing to achieve energy conservation goals. Using the Heihe Jinpen Reservoir as an example shows that when the difference between surface water temperature and that of the lower water column narrows to 3 °C, the system can stop running. At this point, the reservoir can be continuously mixed via the decrease in air temperature. When compared to a situation without artificial intervention (all year round), the mixing temperature of the reservoir increased by nearly 5 °C with artificial mixing. Thus, if there was no artificial intervention, the reservoir mixing temperature was 10 °C; however, during the operation of artificial mixing, this temperature was 15 °C. The microbes in the reservoir maintained high metabolic capacity during the mixing period. This helps to explain the subsequent processes in aeration and mixing that can improve water quality.

As safe water supply sources and energy conservation are of high importance, optimizing the operation of the aeration system warrants further research. This research demonstrates that reservoir managers should consider making full use of the natural mixing in reservoirs caused by decreases in air temperature to keep achieving water quality improvements with low energy costs.

Key water quality and microbial community indicators were studied during the continuous mixing (natural mixing) process in a drinking water reservoir. During the continuous process, water quality had improved whilst pollutant loads in the surface sediments also experienced a variation. Microbial metabolic activity increased with sufficient DO and relatively high water temperature. AWCD values, carbon sources utilization, and the U and H diversity indices also showed an increasing trend during the process. As for surface sediment, TN and OC showed a decreasing trend. RDA analysis indicated that temperature and DO were the key factors affecting changes to the microbial community. The results suggest that the management of water quality is partially dependent on the increase of DO in the reservoir, which promotes increased microbial metabolic activity. This study's results help to reveal the mechanism of TOC removal, giving new insights into the reduction of the precursor of disinfection by-products. It is of great significance for safe drinking water supply.

**Supplementary Materials:** The followings are available online at http://www.mdpi.com/2071-1050/11/24/7106/s1, Figure S1: Water temperature and DO concentrations variations in the water column during the artificial mixing process (1st means after 1 day since the running of the lifting aeration system), Table S1: Critical environmental factors influencing the spatial and temporal variation for the microbial community during the operation of the water-lifting aerators (WLAs).

**Author Contributions:** The co-authors contributed together to the completion of this article. Individual contributions: Experiment and measurement, Z.Z.; conceptualization, T.H.; data curation, W.G.; formal analysis, Y.L. (Yang Li) and Y.L. (Yue Liu); methodology, S.Z.; validation, M.C. All authors read and approved the final manuscript.

**Funding:** This research received no external funding.

**Acknowledgments:** This research was funded by the Key Scientific Research Projects of Higher Education Institutions in Henan Province (20A560023, 19B560012), the Science and Technology Guidance Project of the China Textile Industry Federation (2018040), the Research Team Development Project of Zhongyuan University of Technology (K2018TD004), and the Young Backbone Teachers Grant Scheme of Zhongyuan University of Technology.

**Conflicts of Interest:** The authors declare no conflict of interest.

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
