# Peer review of "Water Quality Responses during the Continuous Mixing Process and Informed Management of a Stratified Drinking Water Reservoir"

_sustainability, doi:10.3390/su11247106_

Round 1
Reviewer 1 Report
Thank you for the opportunity to revise this interesting study.
The water quality of the reservoirs is key for drinking water production, and the studies that investigate this important issue are pivotal to improve the process and to avoid critical health issues.
I have found the manuscript well presented, and the study design appropriate.
The methods and results are well presented and clear.
I have only a few suggestions for the authors, which are listed below.
1) Introduction section. I believe that this section must be improved by the authors, particularly referring to the importance of the reduction of water quality problems (e.g. nitrogen and organic matter). In particular, as the authors correctly underline, one of the major concerns is the presence of organic matter, whose removal from drinking water reservoirs, as stated by the authors in lines 46 and 47, is gaining research attention. In fact, as it has been underlined by different studies, one of the major concerns about the production of drinking water is represented by the disinfectant by-products (DBPs), which can appear after using disinfectants to treat water with a high concentration of organic compounds. The presence of DBPs has affected the water quality in Europe (particularly in Italy), leading to the highest request for derogations from the European quality standards. Moreover, once that derogations were not granted anymore, it was observed, particularly in the regions with the water supply from reservoirs with high starting concentration of the organic compounds (i.e. Sardinia, Italy), that the number of ordinances for limiting the use of drinking water has increased, as well as the concern of the population on the drinking water quality. For these reasons, the authors can focus on these consequences related to the scarce quality of the water reservoirs, in order to strengthen the research question of their study. I suggest the following papers, which can be helpful for this aim:
- Dettori, M.; et al. Qualitative and quantitative aspects of drinking water supply in Sardinia, Italy. A descriptive analysis of the ordinances and public notices issued during the years 2010–2015. Annali di Igiene: Medicina Preventiva e di Comunità 2016, 28, 296–303.
- Azara A, et al. Derogation from drinking water quality standards in Italy according to the European Directive 98/83/EC and the Legislative Decree 31/2001 - a look at the recent past. Ann Ig 2018;30:517-26. doi:10.7416/ai.2018.2252.
- Dettori, M.; et al. Population Distrust of Drinking Water Safety. Community Outrage Analysis, Prediction and Management. Int. J. Environ. Res. Public Health 2019, 16, 1004.
2) Introduction section, line 67. Something happened with the first point of the study purpose, and it is not clear. Please rephrase it.
3) Materials and methods section, lines 73-83. Is there any sampling methods' reference that the authors can refer to? Any standard or law?
4) Materials and methods section. Headings of the paragraphs no. 2.2 and 2.3. Please use Italics.
5) Results and discussion section, figures no. 5 and 6. Please use Italics for the genus and species names within the legends.
6) Results and discussion section, Table no. 3. I have difficulties to understand the table. I suggest the author re-design it.
7) Conclusion section. In light of the importance of the reservoirs' water quality, especially of the presence of the organic compounds (and the consequent DBPs appearance), the authors must improve this section, adding the possible applications of the results obtained, as well as declaring the future implications of their research.
Author Response
I have only a few suggestions for the authors, which are listed below.
1) Introduction section. I believe that this section must be improved by the authors, particularly referring to the importance of the reduction of water quality problems (e.g. nitrogen and organic matter). In particular, as the authors correctly underline, one of the major concerns is the presence of organic matter, whose removal from drinking water reservoirs, as stated by the authors in lines 46 and 47, is gaining research attention. In fact, as it has been underlined by different studies, one of the major concerns about the production of drinking water is represented by the disinfectant by-products (DBPs), which can appear after using disinfectants to treat water with a high concentration of organic compounds. The presence of DBPs has affected the water quality in Europe (particularly in Italy), leading to the highest request for derogations from the European quality standards. Moreover, once that derogations were not granted anymore, it was observed, particularly in the regions with the water supply from reservoirs with high starting concentration of the organic compounds (i.e. Sardinia, Italy), that the number of ordinances for limiting the use of drinking water has increased, as well as the concern of the population on the drinking water quality. For these reasons, the authors can focus on these consequences related to the scarce quality of the water reservoirs, in order to strengthen the research question of their study. I suggest the following papers, which can be helpful for this aim:
- Dettori, M.; et al. Qualitative and quantitative aspects of drinking water supply in Sardinia, Italy. A descriptive analysis of the ordinances and public notices issued during the years 2010–2015. Annali di Igiene: Medicina Preventiva e di Comunità 2016, 28, 296–303.
- Azara A, et al. Derogation from drinking water quality standards in Italy according to the European Directive 98/83/EC and the Legislative Decree 31/2001 - a look at the recent past. Ann Ig 2018;30:517-26. doi:10.7416/ai.2018.2252.
- Dettori, M.; et al. Population Distrust of Drinking Water Safety. Community Outrage Analysis, Prediction and Management. Int. J. Environ. Res. Public Health 2019, 16, 1004.
Reply: Thank you for your comments.
Disinfection by-products are really an important problem in the drinking water. It is also an important reason for the removal of organics from raw water. So in order to explain the necessity of organic matter removal, in the introduction part, some references were added to the manuscript. Especially the important ones that expert supplied. Thank you again.
Dettori, M.; et al. Qualitative and quantitative aspects of drinking water supply in Sardinia, Italy. A descriptive analysis of the ordinances and public notices issued during the years 2010–2015. Annali di Igiene: Medicina Preventiva e di Comunità 2016, 28, 296–303. Azara A., et al. Derogation from drinking water quality standards in Italy according to the European Directive 98/83/EC and the Legislative Decree 31/2001 - a look at the recent past. Ann Ig 2018; 30: 517-26. doi:10.7416/ai.2018.2252. Dettori, M.; et al. Population Distrust of Drinking Water Safety. Community Outrage Analysis, Prediction and Management. J. Environ. Res. Public Health. 2019, 16, 1004.
2) Introduction section, line 67. Something happened with the first point of the study purpose, and it is not clear. Please rephrase it.
Reply: Thank you for your comments.
In the first point, we meant that during the process of continuous mixing, how DO, TN, TP and TOC concentrations varied. Continuous decrease? Or other laws of variation? How does microbial metabolic activity change? In the aspect of summary, there are some problems, which cause trouble for understanding.
We revised it to “during the continuous mixing period, what is the response of TN, TP, TOC, Fe and Mn concentrations in the reservoir”.
3) Materials and methods section, lines 73-83. Is there any sampling methods' reference that the authors can refer to? Any standard or law?
Reply: Thank you for your comments.
Two references were added in this part.
Xuan, L., et al. Effects of rainfall patterns on water quality in a stratified reservoir subject to eutrophication: Implications for management. Sci. Total. Environ. 2015, 521-522: 27-36. Zhou, Z., Huang, T., Li, Y., et al. Sediment pollution characteristics and in situ control in a deep drinking water reservoir. J. Environ. Sci. 2016: 52(2): 223-231.
4) Materials and methods section. Headings of the paragraphs no. 2.2 and 2.3. Please use Italics.
Reply: Thank you for your comments.
Thank you for your rigorous scientific attitude. We have revised them in the manuscript.
5) Results and discussion section, figures no. 5 and 6. Please use Italics for the genus and species names within the legends.
Reply: Thank you for your comments.
Thank you for your rigorous scientific attitude. We have revised them in the manuscript.
6) Results and discussion section, Table no. 3. I have difficulties to understand the table. I suggest the author re-design it.
Reply: Thank you for your comments. We have deleted the Table 3 and moved it to the supplementary materials.
In the manuscript, we added some illustration. “On phylum level, temperature and CHl-a were the most important drivers, which exhibited significant positive correlation with RDA1 and the correlation coefficient reached 0.76(P<0.05) and 0.53 (P<0.05). On genus level, TN (R=0.42, P<0.05), TOC (R=0.47, P<0.05), temperature (R=0.64, P<0.05), DO (R=-0.40, P<0.05) and CHl-a (R=0.54, P<0.05) were key environmental factors, which exhibited significant correlation with RDA1. From the above results, we can see that during the continuous mixing period, the most important physical factors water temperature and DO concentrations were changed significantly. For the improvement of water quality, we should focus on the control of water temperature and DO.”
7) Conclusion section. In light of the importance of the reservoirs' water quality, especially of the presence of the organic compounds (and the consequent DBPs appearance), the authors must improve this section, adding the possible applications of the results obtained, as well as declaring the future implications of their research.
Reply: Thank you for your comments.
We have revised this part, and add “This study results helped reveal the mechanism of TOC removal, giving new insights in the reducing of precursor of disinfection by-products. It was of great significance for safe drinking water supply.” to the conclusion section.

Reviewer 2 Report
The reviewed article deals with the topic of water quality responses during constant mixing processes and management in a stratified drinking water reservoir. As noted by the authors, this is an important issue due to the potential economic benefits resulting from the integration and optimization of continuous and artificial mixing processes, especially in the context of drinking water suitability.
The article was written in a coherent and legible manner, setting four main goals, which were reliably achieved in the part concerning the analysis of results and discussions (goals: evolution of water quality in the reservoir during the continuous mixing period; microbiological activity of organisms in the tank and its utilization of carbon sources; conditions related to nitrogen, organic matter and metals in the microorganism community and finding links between artificial and continuous mixing).
The introduction contains references to reliable scientific sources – articles in peer-reviewed journals of various ranges. The results are presented in a legible way in the form of charts and tables. Appropriate conclusions from the analysis were also drawn. The quality of the presented content is high, as is the scientific level, the attractiveness of the text for readers and the overall substantive value. The originality and significance of the content can be classified as average, however, this is due to the fact that these are preliminary studies that can be the foundation for more extensive analyzes.
The research was planned correctly, the methods were thoroughly described and the conclusions are supported by the presented results. English is at a high level, the phrases used will be understandable even by non-specialists, high clarity has been demonstrated.
The authors should be praised for choosing literature and referring to their own items, which shows that the reviewed article is the result of previous work and is consistent with the achievements of scientists.
No plagiarism was detected, no conflicts of interest were identified, and no ethical contraindications were observed regarding the conducted research.
Despite the generally high level of the article, several elements need to be improved in order for it to be accepted for publication:
1) Correction of typos and some editorial errors:
- line 16 and following - "in situ" instead of "in situ" (italics)
- by line 27 and following - instead of "15 °C" should be "15°C" (without spaces)
- from page 9 to 11 the heading should be corrected – page numbering starts again
- line 296-297 – different font size
2) Presentation of content:
- Figure 1. – sign the points (name them; in the text you can write where in the reservoir they are); the boundaries of the water reservoir should be marked; please add the location of the research site on a larger scale (China, Asia, etc.); please add an arrow (arrows) with the direction of the river flow
- Figure 2. – in the legend it should be clarified what are the positions "1st", "9th" etc. – are these the next days of using the aerator?
- Table 1. – in the upper left part of the table, the cell should be divided in half, with "Water quality indicators" at the top and "Sampling Date" at the bottom (SD \ WQI); please correct the typo in the table signature – "indicators", not "indictors"
- Figure 3. – The same note as in Figure 2. as to the legend
- Figure 5. – "Sampling Time (day)", not "Sanmpling Time (day"); the legend should be italic
- Figure 6. – Latin names of taxa should be in italics
3) Bibliography:
- please add a DOI number to each item (where possible)
- no citations from Sustainability – some items should be cited to show thematic consistency with the journal you are applying to
4) Questions and substantive suggestions:
- Please explain why these and not other measuring points were chosen. What was the frequency of research? How long did it take?
- What is the value of "/31" in line 106 (in equation)? Where did it come from?
- What research do you plan to carry out after these? Do you also anticipate other objects to check the elements that you are examining? I would like a short description of the planned steps.
- The conclusions are supported by the results, however they are laconic – I would advise you to move subchapter 3.7. to applications or enrich applications with at least percentages related to parameter change trends.
Author Response
Despite the generally high level of the article, several elements need to be improved in order for it to be accepted for publication:
1) Correction of typos and some editorial errors:
- line 16 and following - "in situ" instead of "in situ" (italics)
- by line 27 and following - instead of "15 °C" should be "15°C" (without spaces)
- from page 9 to 11 the heading should be corrected – page numbering starts again
- line 296-297 – different font size
Reply: Thank for your comments.
We have revised the errors. For "15 °C", after communicating with the editor, we think the space needs to be kept. Other errors were corrected in the manuscript.
2) Presentation of content:
- Figure 1. – sign the points (name them; in the text you can write where in the reservoir they are); the boundaries of the water reservoir should be marked; please add the location of the research site on a larger scale (China, Asia, etc.); please add an arrow (arrows) with the direction of the river flow
Reply: Thank for your comments.
We have added “Heihe Jinpen Reservoir (34°42′-34°13′N; 107°43′-108°24′E), located in Zhouzhi County, Xi’an City, Shaanxi Province,” to the manuscript. And Figure 1 had been replaced.
|
- Figure 2. – in the legend it should be clarified what are the positions "1st", "9th" etc. – are these the next days of using the aerator?
Reply: Thank for your comments.
This is the sampling date after the lifting aeration system stops running. For example, 9th means after 9 days since the stop of the lifting aeration system, water samples were collected.
In the legend we have added an illustration.
- Table 1. – in the upper left part of the table, the cell should be divided in half, with "Water quality indicators" at the top and "Sampling Date" at the bottom (SD \ WQI); please correct the typo in the table signature – "indicators", not "indictors"
Reply: Thank for your comments.
We have revised the errors.
- Figure 3. – The same note as in Figure 2. as to the legend
Reply: Thank for your comments.
We have revised the errors. In the legend we have added an illustration.
- Figure 5. – "Sampling Time (day)", not "Sanmpling Time (day"); the legend should be italic
Reply: Thank for your comments.
We have revised the errors.
- Figure 6. – Latin names of taxa should be in italics
Reply: Thank for your comments.
We have revised the errors.
3) Bibliography:
- please add a DOI number to each item (where possible)
- no citations from Sustainability – some items should be cited to show thematic consistency with the journal you are applying to
Reply: Thank for your comments.
We checked several published articles. It seems that the article format does not require quotation DOI number. According to your suggestion, we added DOI number to each item in the manuscript.
Articles on Sustainability had been cited in this manuscript, for example reference 3.
4) Questions and substantive suggestions:
- Please explain why these and not other measuring points were chosen. What was the frequency of research? How long did it take?
Reply: Thank for your comments.
These sampling sites were all in the main reservoir. And the depths of the water layer were nearly the same. So we these sampling sites were chosen. The water quality of these sites can basically represent the water quality of the main reservoir.
In this research, water samples were nearly every 10 days. All this research lasted for 47 days. This was because that after the research, the water quality of the reservoir basically remained unchanged
- What is the value of "/31" in line 106 (in equation)? Where did it come from?
Reply: Thank for your comments.
A reference was cited to illustrate the calculation method. In fact, there are 31 carbon sources on each group of eco plate. Average well color development (AWCD) is the average value of microbial utilization of 31 carbon sources. So this is where the 31 came from.
- What research do you plan to carry out after these? Do you also anticipate other objects to check the elements that you are examining? I would like a short description of the planned steps.
Reply: Actually, reservoir water quality improvement methods are of great interest for me. In the process of artificial mixing and natural mixing, I am also interested in the changes of different forms of pollutants. For example, nitrogen is reduced and how different forms are transferred and transformed. Can we do an isotope tracer test in the laboratory? Now I am applying for a research project on nitrogen removal from drinking reservoirs. From the results, N and TOC reduction were observed. So the mechanism was unclear. In Central China, more reservoirs will become urban water sources, and we will face more water quality problems. According to our research, nitrogen and organic matter in these reservoirs are still very important objects of concern. I will gradually take the removal of nitrogen and organic matter in drinking water as my main research direction. Thank you again.
- The conclusions are supported by the results, however they are laconic – I would advise you to move subchapter 3.7. to applications or enrich applications with at least percentages related to parameter change trends.
Reply: Thank you for your comments. We have delete sub chapter 3.7. and enriched the conclusions. Mainly merge subchapter 3.7 and the conclusions.
4. Conclusion
Based on this study and our previous research, we could conclude that aeration and mixing can destroy the stratification of the reservoir and activate the metabolism ability and abundance of indigenous microbes in the reservoir. The water purification mechanism has also been confirmed. When the water quality at the bottom of the reservoir deteriorated and surface algae bloomed, the aeration system started running. When the water quality index reaches a pre-determined target, the system stops running. This is the typical practice in this field of research.
This study demonstrates a new phenomenon, showing that there are improvements to water quality during the continuous mixing process. This study suggests the integration of artificial mixing with natural mixing to achieve energy conservation goals. Using Heihe Jinpen Reservoir as an example shows that when the difference between surface water temperature and that of the lower water column narrows to 3 ℃, the system can stop running. At this point, the reservoir can be continuously mixed via the decrease in air temperature. When compared to a situation without artificial intervention (all year round), the mixing temperature of water reservoir increased by nearly 5 ℃ with artificial mixing. Thus, if there was no artificial intervention, the reservoir mixing temperature was 10 ℃; however, during the operation of artificial mixing, this temperature was 15 ℃. The microbes in the reservoir maintained high metabolic capacity during the mixing period. This helps explain the subsequent processes in aeration and mixing that can improve water quality.
As safe water supply sources and energy conservation are of high importance, optimizing the operation of the aeration system warranted further research. This research demonstrates that reservoir managers should consider making full use of the natural mixing in reservoirs due to decreases in air temperature to keep achieving water quality improvements with low energy costs.
Key water quality and microbial community indicators were studied during the continuous mixing (natural mixing) process in a drinking water reservoir. During the continuous process, water quality had improved whilst pollutant loads in the surface sediments also experienced a variation. Microbial metabolic activity increased with sufficient DO and relatively high water temperature. The AWCD value, carbon sources utilization, as well as the U and H diversity indices also showed an increasing trend during the process. As for surface sediment, TN and OC showed a decreasing trend. RDA analysis indicated that temperature and DO were the key factors affecting changes to the microbial community. The results suggest that the management of water quality is partially dependent on the increase of DO in the reservoir, which promotes increased microbial metabolic activity. This study results helped reveal the mechanism of TOC removal, giving new insights in the reducing of precursor of disinfection by-products. It was of great significance for safe drinking water supply.

Reviewer 3 Report
This study presents water quality improvement during artificial mixing in the Heihe Jinpen Reservoir. Particularly, this study focused on microbial community. I think this study will be useful. However, they need to show physical performance of mixing condition. In other words, I wonder how much aeration is added and how much discharge is added from upstream area or mixing periods.
Comments
Add physical mixing conditions like aeration or DO concentration and discharge quantity, discharge temperature. in Figure 1, add latitude and longitude and flow directions. In Figure 2, where is the monitoring site? in Figure 2, explain 1st day of DO condition. in line 107, add references for Shannon and McIntosh indices.Author Response
Comments
Add physical mixing conditions like aeration or DO concentration and discharge quantity, discharge temperature. in Figure 1, add latitude and longitude and flow directions. In Figure 2, where is the monitoring site? in Figure 2, explain 1st day of DO condition. in line 107, add references for Shannon and McIntosh indices.
Reply: thank you for your comments.
1) According to your valuable suggestion, we supplied “Figure S1. Water temperature and DO concentrations variations in the water column during the artificial mixing process (1st means after 1 day since the running of the lifting aeration system; meaning the sampling date)” in the supplementary materials.
2) As known, in winter, Northwest China is dry and rainless. Therefore, during the study period, the amount of water from upstream is very small, nearly 1-2 m3/s. Compared with the amount of water in the main reservoir, these water volumes have little impact on the main reservoir area. The water temperature of the upstream was lower than that of the main reservoir. During the research period, the water temperature was 10-12 ℃.
3) Latitude and longitude and flow directions had been added in the manuscript: 34°42′-34°13′N; 107°43′-108°24′E. We also replaced Figure 1.
|
4) The values in Figure 2 were the mean values of S1, and S2 and S3. In the manuscript we have added the missing information. DO concentrations on 1st day in Figure 2 showed that DO concentration of the bottom water was 2 mg/L. DO concentration in the upper water layer was high reaching > 7 mg/L. This indicated that the water body was basically in the state of oxygen enrichment. The high DO concentration nearly water depth 75-80 meters was because that the microporous aeration plate of the lift aeration system was at this position.
5) We have added a reference.

Round 2
Reviewer 1 Report
I believe that the scientific quality of the manuscript has increased, and I recommend it for the publication in Sustainability.
Reviewer 3 Report
The authors have revised the manuscript based on the reviewer's comments.
This manuscript is ready to be published in Sustainability Journal.